# Experiences of Patient Safety Education and Factors Affecting the Willingness to Participate in Patient Safety in Undergraduate Nursing Students in South Korea

**DOI:** 10.3390/healthcare12010054

**Published:** 2023-12-26

**Authors:** Sun-Hwa Shin, On-Jeon Baek

**Affiliations:** Nursing Department, College of Nursing, Sahmyook University, Seoul 01795, Republic of Korea; shinsh@syu.ac.kr

**Keywords:** patient safety, nursing students, perception, willingness to participate

## Abstract

Patient safety (PS) education must be provided to medical personnel who deal with life and death situations. This study aimed to examine the current state of PS education and to investigate the factors affecting the willingness of undergraduate nursing students to participate in PS. This study was a cross-sectional descriptive correlational survey. In this study, an online survey targeting members of an online café in South Korea that prepares nursing students was conducted from 19–21 April 2023. In total, 296 undergraduate nursing students in their fourth grade at 132 nursing colleges across the country participated. The results showed that 94.6% (280 students) had experience in PS education and 45.6% (135 students) had witnessed a PS incident in clinical practice. Most students (90.9%) responded that PS education was necessary and that ‘clinical practice at the hospital’ was the most effective method of PS education. PS perception increased as PS knowledge increased, and it was confirmed that an increase in PS perception increased the willingness to participate in PS. Factors that increased nursing students’ willingness to participate in PS were high satisfaction with clinical practice, high PS education needs, and high PS perception scores. To increase nursing students’ willingness to participate in PS, it is important to improve their PS perception and ensure that they have a variety of PS education experiences, which requires the development of a standardized PS education program.

## 1. Introduction

Patient safety (PS) is a fundamental principle of healthcare delivery and a basic human need that must be met [1]. The WHO defines ‘patient safety’ as follows: “A framework of organized activities that creates cultures, processes, procedures, behaviors, technologies and environments in health care that consistently and sustainably lower risks, reduce the occurrence of avoidable harm, make errors less likely and reduce the impact of harm when it does occur” [2]. Patient safety should be a priority in modern healthcare, and policies are being developed to guide clinical practice and education to ensure the universal safety of patients [2]. 

In Korea, problems related to patient safety were not recognized as serious because of the lack of actual data that could estimate the instrument of patient safety incidents [3]. However, the Korea Patient Safety Reporting & Learning System (KOPOS), which was introduced in 2015, identified 52,132 patient safety incidents from 2017 to December 2021, and the number of patient safety incidents reported every year is increasing [4]. This increase in the number of reported patient safety incidents emphasizes the responsibility of preventing patient safety incidents among healthcare providers providing direct patient care [5]. In addition, hospitals are complex spaces with multiple developments every minute, where issues that threaten patient safety cannot be fully predicted; therefore, it is necessary for healthcare workers to cooperate in building a safe medical environment [6]. Because structural systems such as regulations and facilities for patient safety cannot fully guarantee patient safety, it is necessary to improve patient safety activities through various approaches.

Several studies have suggested that healthcare curricula should be redesigned from a patient safety perspective [7,8]. In particular, since nursing students are medical personnel who will enter the hospital field immediately after graduation and will be in charge of practical work through direct contact with patients, it is necessary to acquire the correct knowledge and attitudes about patient safety through the major curriculum [9]. To perform patient safety practices well, it is necessary to systematically educate nursing students on the principles of patient safety so that they can provide safe services to patients.

According to previous research, nursing students want the opportunity to establish safety principles and learn more deeply through the major curriculum [10]. Additionally, nursing students perceived patient safety as important and reported that, as future healthcare providers, they would play an important role in ensuring patient safety [11]. Nursing students’ knowledge and attitudes toward patient safety are positively correlated with performance confidence [12], and nursing students’ patient safety knowledge and attitudes increase their self-efficacy [13]. Since proper knowledge about patient safety positively influence nursing students’ confidence on patient safety in their performance and self-efficacy [12,13], it is necessary to establish proper knowledge about patient safety in the curriculum (both theory and practice) of nursing students.

In the curriculum of nursing colleges in Korea, several major courses deal with the topic of patient safety [14]. The concept of patient safety is partially covered through theoretical and practical education, while the scope of education time and content is limited [8]. There is no separate course labeled “patient safety” in the nursing curriculum [15], and it has been reported that nursing colleges vary in the content and methods of teaching patient safety and courses that include patient safety content [16]. In the four-year nursing college curriculum in Korea, 81.6% of colleges received patient safety education during class, and the contents of patient safety education conducted for nursing students included safety incident content and coping, communication, clinical safety, safety culture, teamwork and cooperation, and safety systems [14]. Patient safety education conducted for the general public of inpatients included patient safety activities, such as patient identification, fall prevention, infection prevention, pressure ulcer prevention, fire prevention, and accurate communication [17]. Nursing students directly experience patient safety incidents during clinical practice [18]. Therefore, it is necessary for medical institutions to check the contents of patient safety education for nursing students before implementing it in clinical practice and to provide regulations and operating policies regarding patient safety [14]. Efforts should be made to bridge the gap between theory and practice through open communication with medical institutions so that patient safety education emphasized in practice can be reflected in the curriculum of nursing colleges.

Providing teaching and training in the concepts and principles of patient safety to nursing students who will grow into future nurses is essential to realize safe nursing required in clinical practice. According to a previous study of nursing professors, patient safety was covered in various major subjects, such as fundamental nursing, adult nursing, and nursing management through lectures and practical exercises. It was reported that each university operates autonomously because appropriate standards for patient safety education are not specified [14]. As recent studies have investigated the educational needs of nursing students regarding patient safety [19,20], there is a need to develop practical educational programs for nursing students. Therefore, this study aims to examine the contents of patient safety education targeting nursing students and to identify the factors that affect their willingness to participate in patient safety. Through this, it can provide basic data for the development of a patient safety education program for nursing students.

This study examined the content of PS education for nursing students and explored its impact on their willingness to participate in PS. First, the general characteristics of nursing students, the status of PS education (PS education experience, witnessed PS incident, PS education needs, PS education timing, and effective training method), and major subjects dealing with PS were identified. Second, the relationship among nursing students’ PS knowledge and perception and willingness to participate in PS was confirmed. Third, the effects of the general characteristics of nursing students, PS education status, PS knowledge, and PS perception on willingness to participate in PS were confirmed.

## 2. Materials and Methods

### 2.1. Study Design and Setting

This study was a cross-sectional descriptive correlational survey targeting undergraduate nursing students. This was conducted to confirm the content of PS education experienced by nursing students and to examine its impact on their willingness to participate in PS.

### 2.2. Research Participants

This study targeted undergraduate nursing students enrolled in their fourth year of nursing in South Korea. The eligibility criteria were nursing students who were members of the online café (a group that prepares nurses), enrolled in a fourth-year nursing program, had experience in clinical practice for at least two semesters, completed major courses in the nursing program, were able to read and respond to the online questionnaire on their own, and voluntarily agreed to participate in the study. The exclusion criteria included students who were currently on a leave of absence, had graduated, or had limitations regarding reading and responding to the questionnaire. At the beginning of the survey, questions were asked to indicate whether the participants met the eligibility criteria (university, grade, completion of two or more semesters of clinical practice, etc.), and if the eligibility criteria were met, the survey was conducted. The survey was set to end when the first 300 people completed it.

Using the G*power program (version 3.1.9.7) to calculate the sample size, the minimum number of cases was 194 based on a regression analysis with a moderate effect size of 0.15, a one-type error of 0.05, a power of 0.95, and 14 independent variables (gender, age, satisfaction with major, clinical practice period, satisfaction with clinical practice, experience of PS education, types of PS incidents, witnessed PS incident, needs for PS education, PS education timing, effective training method, PS knowledge, PS perception, and PS willingness). The study included 300 nursing students enrolled in 132 nursing colleges across the country; the data from 296 students were analyzed after removing four students who had no clinical practice experience due to COVID-19. Thus, the criterion for the minimum number of cases was satisfied.

### 2.3. Research Instruments

#### 2.3.1. PS Knowledge

PS knowledge was measured using the Patient Safety Management Knowledge Scale (PSKS) developed by Choi and Lee [21]. The PSKS was developed based on the contents of patient safety assessment items related to nursing activities from previous studies [12,22]. The PSKS consists of 10 questions, each of which requires a ‘yes’, ‘no’, or ‘don’t know’ response. Correct answers were scored as 1, and incorrect answers and ‘don’t know’ were scored as 0. Scores ranged from 0 to 10, with higher scores indicating greater PS knowledge. In the study of Choi and Lee [21], the Cronbach’s α of the instrument was 0.65, and in this study, it was 0.58.

#### 2.3.2. PS Perception

PS perception was measured using the Perception of Importance on Patient Safety Management Scale (PI-PSM) developed by Park et al. [1] and modified and supplemented for nursing students by Choi and Lee [21]. Park et al. [1] developed the PI-PSM to evaluate the perception of the importance of patient safety management among healthcare workers, and Choi and Lee [21] modified the instrument to measure the perception of patient safety management among nursing students. The PI-PSM consisted of 21 questions and was composed of 4 sub-factors (concern about PSM, confidence about PSM, will for PSM, and recognition about PSM) [1]. On the other hand, Choi and Lee [21] modified the questionnaire to 10 questions suitable for nursing students through content validity by experts, and they did not perform construct validity. The PS perception consists of 10 items, with responses on a 5-point Likert scale ranging from 1 point for “not at all” to 5 points for “very much so”. The scores range from 10 to 50, with higher scores indicating a more positive perception toward patient safety. In the study of Choi and Lee [21], the Cronbach’s α of the instrument was 0.67, and in this study, it was 0.65.

#### 2.3.3. Willingness to Participate in PS

Willingness to participate was measured using the Willingness to Participate in Patient Safety Scale (WPPSS) developed by Lee [23]. The WPPSS was developed based on a literature review of 20 guidelines for preventing medical errors published by the Agency for Healthcare Research and Quality (AHRQ) in the United States, and the questions were validated for content validity by experts [23]. The questionnaire consisted of 18 questions and five sub-factors (one question for decision making, three questions for information provision, six questions for asking questions, five questions for confirming, and three questions for reporting). Each question is measured on a 4-point scale, with 4 points for “very much”, 3 points for “agree”, 2 points for “no”, and 1 point for “not at all”. A higher score indicates a greater willingness to participate in patient safety. In a previous study by Lee [23], the Cronbach’s α of the instrument was 0.88, and in this study, it was 0.92.

### 2.4. Data Collection and Ethical Considerations

This study deliberated on the research procedures and ethical considerations at the Institutional Review Board of Sahmyook University to which the researchers belong (IRB No: SYU 2023-01-006-002). After submitting to a research ethics review, data were collected from 19 April to 21 April 2023 online surveys. To recruit nursing students, we contacted the café manager, asked for permission, and posted a notice in the fourth-year meeting room. The questionnaire took approximately 10 min to complete. Before conducting the online survey, we posted an informed consent statement introducing the purpose, content, and study procedures. It was explained that the participants were free to stop answering the survey and withdraw their consent and that there would be no disadvantages. Written consent was obtained after reading the explanation and checking the ‘consent’ box on the online questionnaire. In addition, in order to protect the anonymity and privacy of the participants who participated in the study, it was suggested that the collected data, such as university, age, gender, mobile phone number, be stored in a locked file in the researcher’s computer folder; incomplete data were stored in a locked file, which was set to be automatically deleted, and information about this was included in the research description. It was specified that the data collected thereafter would be stored for 3 years and then discarded. Participants’ mobile phone numbers were collected and a coupon of 3000 won (KRW) was given to those who completed all surveys.

### 2.5. Data Analysis

Data analysis was performed using the SPSS program (version 25.0; IBM Corporation, Armonk, NY, USA). The mean, standard deviation, frequency, and percentage of the participants’ general characteristics and PS education content were calculated. The mean, standard deviation, skewness, and kurtosis of PS knowledge, perception, and willingness to participate were calculated. The differences in PS knowledge, perception, and willingness to participate according to the general characteristics of nursing students were analyzed via *t*-test and ANOVA, and Pearson’s correlation coefficient was used to determine correlation. Finally, hierarchical regression was performed in three steps. Through this, the impact of general characteristics, PS knowledge, and PS perception on willingness to participate was confirmed. All analyses had a statistical significance level of 0.05 or less.

## 3. Results

### 3.1. Participant Characteristics

The general characteristics of the study participants are presented in Table 1. As for the location of nursing colleges, Gyeongsang-do had the highest at 24.7% (73 students), and Gangwon-do had the lowest at 9.1% (27 students). As for gender, females accounted for the majority at 91.9% (272 students). Among the participants, 55.4% (164 students) were aged 20–22, with a mean age of 23.1 years. Regarding major satisfaction, ‘satisfaction’ was 78.0% (231 students), indicating that most students were highly satisfied with their majors. As for the period of clinical practice, ‘more than 13 weeks’ accounted for the most at 73.0% (216 students), and ‘satisfaction’ showed the largest distribution at 68.9% (204 students) for clinical practice satisfaction.

### 3.2. Patient Safety Education Content

The content of PS education in the curricula conducted at nursing colleges located nationwide was also investigated (Table 2). The majority of students (280 (94.6%)) reported that they had experience in PS education, while 16 (5.4%) reported that they had no such experience. In response to subjects dealing with patient safety, fundamental nursing was the most common at 76.0% (225 students), and most major subjects covered PS in theory and practice. In most nursing colleges, the theory and practice of major courses dealt with the concepts of ‘patient safety’, patient identification, hand hygiene, medication, infection control, falls, and pressure sores. PS incidents were reported by 54.4% (161 students) of the participants who did not witness them and 45.6% (135 students) of the participants who did. When the PS incidents that were witnessed by the students (n = 135) were investigated, the most common type was falls at 30.7% (91 students), followed by injection and medication errors at 4.1% (12 students), patient identification errors at 2.7% (8 students), and needle stick incidents at 2.4% (7 students). Regarding PS education needs, 90.9% (269 students) of the students answered ‘Necessary’, which accounted for the majority. Regarding PS education needs, 90.9% (269 students) of the students answered ‘Necessary’, which again accounted for the majority. Regarding the time of PS education, 46.6% (138 students) of the second-year students who are learning the basics of their major accounted for the most, followed by 34.8% (103 students) of the first-year freshmen. The most effective training method was practical training at hospitals (50.3%, 149 students), followed by simulation training (26.7%, 79 students), training using media (video, virtual reality, and augmented reality) (13.9%, 41 students), and handouts and lectures (9.1%, 27 students).

### 3.3. Differences in Patient Safety Knowledge, Perception, and Willingness to Participate by Demographic

The differences in the main study variables according to the participants’ general characteristics are shown in Table 3. PS knowledge differed significantly based on PS education experience (t = −2.80, *p* = 0.005). PS perception differed significantly according to clinical practice satisfaction (t = −3.30, *p* = 0.002), PS education experience (t = −2.65, *p* = 0.008), PS education needs (t = −2.25, *p* = 0.025), and PS education timing (F = 4.78, *p* = 0.003). Post hoc testing of the timing of PS training showed that training in the first, second, and third grade resulted in higher PS perception scores than training in the fourth grade. Willingness to participate in PS was significantly related to gender (t = −2.25, *p* = 0.025), age (t = 3.49, *p* < 0.001), satisfaction with major (t = −2.66, *p* = 0.008), satisfaction with clinical practice (t = −3.72, *p* < 0.001), experience with PS education (t = −2.53, *p* = 0.023), witnessing PS incidents (t = −2.45, *p* = 0.015), and PS education needs (t = −3.79, *p* < 0.001).

### 3.4. Correlations between Patient Safety Knowledge, Perception, and Willingness to Participate

Descriptive statistics and correlations between the main research variables were examined (Table 4). PS knowledge had a mean of 3.47 points (±0.88), PS perception was 39.56 points (±4.99), and willingness to participate in PS was 79.99 points (±9.02). Correlations among PS knowledge, perception, and willingness to participate were examined. PS knowledge was positively correlated with PS perception (r = 0.21 *p* < 0.001). There was no significant correlation between PS knowledge and willingness to participate (r = 0.10, *p* = 0.119). PS perception was positively correlated with willingness to participate in PS (r = 0.42, *p* < 0.001).

### 3.5. Influence of Patient Safety Knowledge and Perception on Willingness to Participate

The underlying assumptions were checked before performing the hierarchical regression. The Durbin–Watson value was found to be 2.15, and the variance inflation factor (VIF) values ranged from 1.05 to 1.28, which satisfied the assumption because there was no problem with the autocorrelation and multicollinearity of the independent variables.

First, in Model 1, which controlled for demographic characteristics, we entered the gender and age and selected two variables that showed significant differences in both the independent and dependent variables as confounders. As a result of the analysis, nursing students’ age, satisfaction with clinical practice, satisfaction with PS education, and PS education needs were found to have a significant effect on willingness to participate in PS, and the explanatory amount was found to be 19.7% (R^2^ = 0.197, F = 8.77, *p* < 0.001). Second, Model 2, in which PS knowledge was an input after controlling for general characteristics, increased the amount of explanation by 0.2% to 19.9% (R^2^ = 0.199, F = 7.89, *p* < 0.001), and PS knowledge had no significant effect on the willingness to participate in PS (β = 0.05, *p* = 0.366). Finally, Model 3 showed that after controlling for the general characteristics and PS knowledge, the addition of PS perception was significant, increasing the amount explained by 10.4% to 30.3% (R^2^ = 0.303, F = 12.36, *p* < 0.001). In the final model, the group with a high satisfaction with clinical practice (β = 0.11, *p* = 0.042), the group with PS education experience (β = 0.14, *p* = 0.008), the group with a need for PS education (β = 0.13, *p* = 0.014), and the group with a more positive PS perception (β = 0.34, *p* < 0.001) had significant effects on the willingness to participate in PS (Table 5).

## 4. Discussion

This study examined the content of PS education among Korean undergraduate nursing students and analyzed their PS knowledge, perception, and willingness to participate. Based on these findings, the following conclusions were drawn.

In this study, most nursing students (94.6%) had experience with PS education. Most majors cover PS education, with fundamental nursing being the most prevalent. The contents of PS education that nursing students responded to were patient identification, medication administration, infection control, falls, and pressure sores. Previous studies have reported that PS education should include accurate training in nursing skills and in handling various instruments [19]. These components of PS must be included in nursing skills so that they can be used in actual nursing situations. Nursing students had a high demand for PS education (90.9%) and responded that clinical practice education in hospitals was the most effective learning method. The next most effective learning method was simulation (26.7%), which is designed around actual nursing scenarios and is a method that allows nursing students to practice in an online environment similar to the real thing [24]. In addition, it is implemented in the form of self-study and is presented as a teaching method that effectively improves the learning performance of nursing students by providing customized feedback to each student [25]. It was confirmed that students considered the simulation learning method, which involves learning in an environment similar to the real thing through direct observation of clinical practice and virtual scenarios, as the most effective learning method. A study of the PS curriculum among fourth-year nursing students found that 81.6% of students received PS education, but their perceived PS competency was only moderate [16]. A study of nursing college professors found that lecture-based instruction predominated at 38.9% and that PS topics were often covered in nursing management courses [26]. In addition, in a systematic literature review of PS education, 81.6% of nursing students reported that they received PS education in class and lecture-type education more often than in practice or simulation classes [14]. Another study reported that nursing students gained more confidence regarding PS in theory classes than in clinical practice [15]. However, in this study, nursing students recognized that clinical practice education was more effective than lecture-type education in learning PS education content, and these results are different from those of previous studies. Based on the results of this study, it is necessary to explore PS education methods for field instructors in clinical practice in addition to lecture-based education in theoretical classes. Currently, the nursing college curriculum deals with the concept of PS through various major courses. However, nursing students from six universities mentioned the name of a separate PS course. In Korea, the accreditation evaluation standards for medical institutions present five core indicators to ensure patient safety: accurate patient identification, accurate communication among medical staff, accurate performance of surgery/procedures, fall prevention activities, and hand hygiene [27]. For this purpose, it is necessary to develop a standardized PS education program that reflects the core indicators of PS through a systematic linkage with clinical practice and to apply it consistently in nursing colleges nationwide.

In this study, 45.6% of the nursing students reported having witnessed a PS incident. The most common types of PS incidents were falls (30.7%), medication errors (4.1%), patient identification errors (2.7%), and needle sticks (2.4%). In a previous study of 228 fourth-year nursing students, 62 students (27.1%) experienced PS incidents, and the types of incidents were reported as follows: falls (50.0%), needle sticks (18.5%), patient identification errors (14.8%), injection errors or medication errors (12.0%), medical device damage (4.3%), and contamination of disinfectant items (3.3%) [18]; these results are similar to the results of this study, where ‘falls’ accounted for a large proportion of PS incidents. The percentage of nursing students who experienced PS incidents during clinical practice was 44.0% in the study by Huh and Kang [28], 44.0% in the study by Song and Moon [29], and 45.1% in the study by Cho [30], which is similar to or less than the distribution in this study. Even if nursing students experienced a PS incident during clinical practice, most could not participate in the process of handling the incident after it occurred [30]. Nursing students in the third and fourth grades of clinical practice had lower scores for the clear expression of opinions when faced with PS incidents than those in the lower grades [15]. It was interpreted that nursing students had difficulty reporting to medical personnel or professors in charge because of the fear of being negatively evaluated when performing in clinical practice [15,28]. Therefore, it is necessary to establish an educational environment in which nursing students can voluntarily report to medical staff or professors immediately when they witness a PS incident, and confidentiality and unfavorable measures are not performed [15]. In addition, to actively and efficiently cope with PS incidents when becoming a nurse, it is necessary to conduct training to participate in the voluntary reporting of PS incidents and the analysis and improvement activities of incidents in the undergraduate curriculum of nursing students.

The nursing students’ willingness to participate in PS was 79.99 points (mean was 4.44). In a study targeting patients, the willingness to participate in PS was 3.45 points [23], and the general public’s willingness to participate in PS was 3.34 points [31]; the willingness to participate among nursing students in this study was high. In this study, the characteristics that made a significant difference in nursing students’ willingness to participate in PS were gender, age, satisfaction with major, satisfaction with clinical practice, experience with PS education, witnessing a PS incident, and PS education needs. The willingness to participate in PS emphasizes an active role in preventing PS incidents and blocking errors before they reach patients or cause harm [23]. There is no study investigating the willingness to participate among nursing students, so direct comparisons are difficult. In previous studies, the willingness to participate in PS increased when the general public and patients had experience with PS education [31,32]. In this study, the higher the degree of satisfaction with major, the more significant the nursing student’s willingness to participate in PS, which is thought to have affected the nursing student’s role in patient safety if their satisfaction with nursing major is high. In addition, nursing students learn patient safety content through various major subjects, but it was found that witnessing PS incidents and direct experience with PS education play major roles in increasing the willingness to participate.

In this study, nursing students’ PS knowledge did not directly affect their willingness to participate in PS, whereas PS perception had a significant effect on their willingness to participate. Previous studies have shown that PS knowledge does not directly influence PS practices [21], which is consistent with our findings. On the other side, a previous study [33] showing that knowledge of medical information increases the willingness to participate in PS was inconsistent with the results of this study. The higher the PS knowledge of nursing students, the higher their awareness of the importance of PS management [21], and the higher their PS perception, the higher their confidence in their PS performance [18]. Even if an increase in knowledge through PS education does not directly translate into a change in behavior, external environmental factors must be considered to translate knowledge into performance. The results showing that nursing students’ PS perception directly affects their willingness to participate in PS suggests that integrated education and training that forms the right perception of PS, rather than education that simply increases knowledge, is needed. It has been reported that it is effective to expose patients to the concept of PS in the lower grades of the undergraduate course and to confirm the concept of PS as the grades progress [34]. In this study, many students responded that it was appropriate to conduct PS education in the second and first year, when learning the basics of the major. It is necessary to organize educational content according to the educational level of nursing students by grade, to explore various learning methods, and to conduct repeated learning so that the accumulated PS perception can be connected to practical behavior. Therefore, designing a curriculum that systematically teaches PS concepts in the lower grades and improves PS perception through practical experiences will increase willingness to participate. It is also necessary for nursing students to apply participatory education through various activities, in addition to theoretical PS education. In addition, future studies should be conducted to identify the various factors that affect nursing students’ willingness to participate in PS, and efficient educational methods should be developed to encourage active participation in PS activities.

This study is meaningful because it identified PS education content learned in the nursing college curriculum by fourth-year nursing students nationwide and analyzed the factors affecting their willingness to participate in PS. Nevertheless, several research limitations exist and improvements are required. First, a nationwide randomized survey of nursing students was conducted; however, the sample size was insufficient and not representative of the regions and classes of the nursing colleges. In addition, since it was limited to café members, there is a limit to generalizing the research results; therefore, a repeat study is suggested to secure the validity and representativeness of the future sampling process. Second, the reliability of the PS knowledge and perception instruments used in this study was low, limiting the validity of the results. Therefore, it is necessary to develop a PS knowledge and perception instrument that secures reliability and validity for nursing students. In addition, we propose a study to measure the level of PS perception of nursing students to determine the impact of PS activities on their practice and performance as clinical nurses after graduation. Third, there is no objective and standardized instrument to investigate the educational experience of patient safety, so we investigated the content covered in theoretical and practical courses. We recommend a study to evaluate the appropriateness of PS education content and organization in the curriculum of each nursing college and to develop a standardized education program that reflects the PS content needed in the clinical field.

## 5. Study Implications

In terms of nursing research, this study provided a basis for understanding the relationship between factors affecting the willingness to participate in PS, as there is a lack of research in South Korea that identifies factors affecting the willingness to participate in PS among nursing students. Based on this, it has provided basic data that can be used in research related to patient safety conducted on nursing students.

In terms of nursing education, nursing students’ PS perception was a factor affecting their willingness to participate in PS, and PS knowledge did not affect their willingness to participate. Based on these findings, we propose the development of an educational program that increases the perception of the importance of PS, rather than education that simply increases knowledge, in order to increase nursing students’ willingness to participate in PS. Rather than lightly covering patient safety topics in various major courses in the Department of Nursing, there is a need to develop a standardized patient safety curriculum that reflects key indicators of patient safety in clinical settings.

In terms of nursing practice, it has been proven that nursing students’ clinical practice satisfaction and PS education experience are influential factors that affect their willingness to participate in PS. Based on this, it is necessary to strengthen practical education that allows nursing students to directly participate in patient safety activities at various practice sites to increase their satisfaction with clinical practice.

## 6. Conclusions

This study identified the educational experiences in PS of fourth-year nursing students and examined the relationship between PS knowledge, perception, and willingness to participate. It was found that the perception of the importance of PS, rather than PS knowledge, was the most powerful factor in increasing the willingness to participate. To increase the willingness to participate in PS among nursing students, it is important to form the right perception toward PS, and for this purpose, the development of a standardized PS education program is required. Through standardized program development, clinical adaptability can be improved by strengthening theoretical and practical education focusing on PS concepts in nursing college curricula. 

## Figures and Tables

**Table 1 healthcare-12-00054-t001:** General characteristics of participants (N = 296).

Characteristics	Categories	n (%)	Mean ± SD
Residential area	Gyeongsang-do	73 (24.7)	
	Gyeonggi-do/Incheon	55 (18.6)	
	Jeolla-do/Jeju-do	52 (17.6)	
	Seoul	47 (15.9)	
	Chungcheong-do	42 (14.2)	
	Gangwon-do	27 (9.1)	
Gender	Female	272 (91.9)	
	Male	24 (8.1)	
Age (year)	20–22	164 (55.4)	23.10 ± 2.71
	≥23	132 (44.6)	
Satisfaction with major	Satisfied	231 (78.0)	
	Unsatisfied	65 (22.0)	
Clinical practice period (month)	More than 4	216 (73.0)	
	Less than 3	80 (27.0)	
Satisfaction with clinical practice	Satisfied	204 (68.9)	
	Unsatisfied	92 (31.1)	

**Table 2 healthcare-12-00054-t002:** Contents of patient safety education (N = 296).

Characteristics	Categories	n (%)
PS education experience	No	16 (5.4)
	Yes	280 (94.6)
Witnessed PS incident	No	161 (54.4)
	Yes	135 (45.6)
Types of PS incidents (n = 135)	Fall	91 (30.7)
	Medication error	12 (4.1)
	Patient identity check error	8 (2.7)
	Injection error	7 (2.4)
	Oral medication error	6 (2.0)
	Contamination	3 (1.0)
	Medical equipment malfunction	3 (1.0)
	Others	5 (1.7)
PS education needs	Necessary	269 (90.9)
	Unnecessary	27 (9.1)
PS education timing	First grade	103 (34.8)
	Second grade	138 (46.6)
	Third grade	39 (13.2)
	Fourth grade	16 (5.4)
Effective training method	Clinical practice at hospital	149 (50.3)
	High-fidelity simulation	79 (26.7)
	Media, VR/AR	41 (13.9)
	Handout, Lecture	27 (9.1)

PS = patient safety; VR = virtual reality; AR = augmented reality.

**Table 3 healthcare-12-00054-t003:** Differences in patient safety knowledge, perception, and willingness to participate according to demographic (N = 296).

Characteristics	Categories	PS Knowledge		PS Perception		Willingness to Participate in PS	
		M ± SD	t/F(*p*)	M ± SD	t/F(*p*)	M ± SD	t/F(*p*)
Gender	Male	3.25 ± 1.22	−1.56 (0.209)	39.08 ± 5.58	−0.49 (0.623)	76.04 ± 10.38	−2.25 (0.025)
	Female	3.49 ± 0.84		39.61 ± 4.95		80.34 ± 8.82	
Age (year)	20–22	3.42 ± 0.82	−0.99 (0.322)	39.46 ± 4.92	−0.41 (0.682)	81.65 ± 7.66	3.49 (<0.001)
	≥23	3.52 ± 0.94		39.70 ± 5.10		77.93 ± 10.12	
Satisfaction with major	Satisfied	3.47 ± 0.84	−0.21 (0.835)	39.86 ± 4.97	−1.91 (0.057)	80.72 ± 8.16	−2.66 (0.008)
	Unsatisfied	3.45 ± 1.02		38.52 ± 4.99		77.38 ± 11.25	
Clinical practice period (month)	Less than 3	3.53 ± 0.89	0.70 (0.484)	39.99 ± 5.28	0.89 (0.376)	79.04 ± 8.29	−1.16 (0.246)
	More than 4	3.44 ± 0.88		39.41 ± 4.89		80.34 ± 9.27	
Satisfaction with clinical practice	Satisfied	3.50 ± 0.78	−0.84 (0.401)	40.20 ± 4.82	−3.30 (0.002)	81.40 ± 7.98	−3.72 (<0.001)
	Unsatisfied	3.40 ± 1.07		38.16 ± 5.11		76.87 ± 10.36	
PS education experience	No	2.88 ± 1.20	−2.80 (0.005)	36.38 ± 4.77	−2.65 (0.008)	71.69 ± 13.76	−2.53 (0.023)
	Yes	3.50 ± 0.85		39.75 ± 4.95		80.46 ± 8.46	
Witnessed PS incident	No	3.38 ± 0.84	−1.88 (0.062)	39.39 ± 5.28	−0.67 (0.501)	78.83 ± 9.36	−2.45 (0.015)
	Yes	3.57 ± 0.92		39.78 ± 4.63		81.38 ± 8.42	
PS education needs	Necessary	3.48 ± 0.85	−0.65 (0.523)	39.77 ± 4.91	−2.25 (0.025)	80.61 ± 8.61	−3.79 (<0.001)
	Unnecessary	3.33 ± 1.14		37.52 ± 5.43		73.85 ± 10.70	
PS education timing	First grade ^a^	3.44 ± 0.79	1.54 (0.203)	40.09 ± 5.05	4.78 (0.003) a,b,c > d	81.55 ± 8.19	2.40 (0.068)
	Second grade ^b^	3.54 ± 0.88		39.54 ± 4.77		79.64 ± 8.40	
	Third grade ^c^	3.44 ± 0.91		40.05 ± 4.36		77.18 ± 12.03	
	Fourth grade ^d^	3.06 ± 1.24		35.19 ± 6.19		79.75 ± 9.66	
Effective training method	Clinical practice	3.37 ± 0.79	0.73 (0.537)	39.04 ± 5.91	1.92 (0.127)	79.30 ± 9.13	0.78 (0.505)
	Simulation	3.48 ± 0.79		39.78 ± 4.59		79.95 ± 9.05	
	Media, VR/AR	3.32 ± 0.96		38.00 ± 4.69		78.54 ± 9.98	
	Handout, Lecture	3.54 ± 1.01		40.15 ± 5.44		81.06 ± 8.42	

PS = patient safety; Note: a, b, c and d refer to post hoc test results using Scheffé’s method.

**Table 4 healthcare-12-00054-t004:** Descriptive statistics and correlation of patient safety knowledge, perception, and willingness to participate (N = 296).

Variables	PS Perception	Willingness to Participate	Mean ± SD	Skewness	Kurtosis
	r (*p*)				
PS knowledge	0.21 (<0.001)	0.10 (0.119)	3.47 ± 0.88	−0.67	1.99
PS perception		0.42 (<0.001)	39.56 ± 4.99	−0.08	−0.60
Willingness to participate			79.99 ± 9.02	−1.16	1.89

PS = patient safety.

**Table 5 healthcare-12-00054-t005:** Influence of patient safety knowledge and perception on willingness to participate (N = 296).

Variables (Reference)	Model 1				Model 2				Model 3				
	b	SE	β	*p*	b	SE	β	*p*	b	SE	β	*p*	VIF
Gender (Male)	1.64	1.87	0.05	0.382	1.47	1.88	0.04	0.434	1.31	1.76	0.04	0.456	1.16
Age (20-22)	−2.78	1.03	−0.15	0.007	−2.89	1.04	−0.16	0.006	−3.10	0.97	−0.17	0.002	1.17
Satisfaction with major (Less than)	0.58	1.28	0.03	0.652	0.60	1.28	0.03	0.643	0.54	1.20	0.02	0.653	1.25
Satisfaction with practice (Less than)	3.24	1.15	0.17	0.005	3.20	1.15	0.16	0.006	2.22	1.09	0.11	0.042	1.28
PS education experience (No)	7.14	2.14	0.18	0.001	6.82	2.17	0.17	0.002	5.45	2.04	0.14	0.008	1.07
Witnessed PS incident (No)	1.70	0.98	0.09	0.084	1.59	0.99	0.09	0.107	1.54	0.92	0.09	0.095	1.06
PS education needs (No)	5.03	1.69	0.16	0.003	4.98	1.69	0.16	0.003	3.93	1.58	0.13	0.014	1.05
PS knowledge					0.51	0.56	0.05	0.366	−0.11	0.53	−0.01	0.840	1.10
PS perception									0.62	0.09	0.34	<0.001	1.13
R^2^ (ΔR^2^)	0.197				0.199 (0.002)				0.303 (0.104)				
F(*p*)	8.77 (<0.001)				7.89 (<0.001)				12.36 (<0.001)				

PS = patient safety; note: adjusted for gender, age, satisfaction with major, satisfaction with clinical practice, PS education experience, witnessed PS incident, and PS education needs.

## Data Availability

The datasets used and/or analyzed in the current study are available from the corresponding author on reasonable request.

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
