# Peer review of "Experiences of Patient Safety Education and Factors Affecting the Willingness to Participate in Patient Safety in Undergraduate Nursing Students in South Korea"

_healthcare, 2023, doi:10.3390/healthcare12010054_

Round 1

Reviewer 1 Report

Comments and Suggestions for Authors

Thank you for your submission. I have a few comments on the paper.

·      The title and the research aim appear not congruent.

·      Consider changing the title to align it with the study purpose.

·      Under content of PS education, I was wondering how the authors assess ‘experience’ of PS education. This is something that is difficult to assess given the nature of the study design.

·      It is also interesting to note that only 4th year students were included in the study. Why students in other levels were omitted from the study?

·      Under the discussion section, the author fails to present and discuss the results related to factors affecting willingness to PS including age, satisfaction with clinical practice, experience with PS, needs for PS education. Additionally, the author provided shallow discussion on the effect of safety attitudes on their willingness. 

·      Also, under recommendations, it is vital to align it with the study result. For instance, the author recommended to increase PS knowledge to increase willingness for PS; however, in the regression analysis, knowledge had no effect on their willingness. 

·      Authors should consider identifying tangible implications for nurse education based on the study result. 

Reviewer 2 Report

Comments and Suggestions for Authors

Thank you for giving opportunity to review this paper. Below are my comments:

1) Change this “This study was a cross-sectional descriptive survey” to” This study was a cross-sectional, descriptive-correlational study” as the authors are studying relationships in addition to describing the variables.

2) Suggest to change the title “2.1. Study design” to “2.1. Study design and setting”; thus, providing more details about the location in which the study was conducted.

3) What is the sampling strategy used to reach interested and eligible participants?

4) When estimating the sample size, the authors mentioned “14 independent variables”. Please write these 14 independent variables in (…) to facilitate understanding.

5) The decrease in Cronbach's α in the PSKS from .65 in the original instrument to .58 in this study raises concerns about the internal consistency reliability of the tool. It is important for the authors to thoroughly address this decrease and justify its use in the study. They also need to address that in the ‘study limitations’ section.

6) Same comment (#4) to the use of PSAS. More details as stated are needed.

7) The authors did a good job in the ‘2.4 Data collection and ethical considerations.’ However, one missing point to ask is whether any measures were taken to ensure the anonymity of participants during the data collection process. It would be important to address whether or how the researchers protected the confidentiality of participants' personal information, such as their mobile phone numbers, to fully comply with ethical considerations.

8) In '2.5 data analysis', it is recommended to change "The average" to "the mean" to maintain consistency in statistical terminology.

9) Could you please provide information on how missing data were handled in the analysis? Also, justify your selection to the “hierarchical regression analysis”

10) It would be helpful to know the total number of nursing students who were initially approached or invited to participate in the study in order to accurately determine the response rate. Plz add the response rate, if available.

11) In the Discussion section, it is essential to provide additional details about the scoring of simulation in Table 2 and clarify the type of simulation that was conducted. It would be beneficial to elaborate on the nature and purpose of the simulation in the context of clinical practice at hospitals. By explaining this further, readers can gain a better understanding of how simulation activities were incorporated into the study and their relevance to nursing education and training.

12) What statistical test was used for Table 3 findings? t-test or ANOVA? If so, please explain that in the "2.5 Data analysis".

13) As “no study is perfect”, authors should explain potential limitations in the “Study limitations” section before the conclusions.

14) This research seems original but how can the authors make a significant contribution to the field? Please add another section called “Study implications” to address your results importance to nursing administration, practice, research, or education.

15) The “Conclusions” section seems too long. It would be helpful to avoid unnecessary repetition or excessive detail in the conclusion, focusing on the main takeaways from the study and their significance for the field of nursing. Also, ending with a recommendation for future research in the field.

16) There are some old references. If the authors are able to cite more recent references as possible, it would be better.

Round 2

Reviewer 2 Report

Comments and Suggestions for Authors

The authors addressed all comments in the revised paper. No further edits